# Integrating eye gaze into machine learning using fractal curves

**Robert Ahadizad Newport**                    ROBERT.NEWPORT@HDR.MQ.EDU.AU
**Sidong Liu**                                 SIDONG.LIU@MQ.EDU.AU
**Antonio Di Ieva**                            ANTONIO.DIIEVA@MQ.EDU.AU
*Computational NeuroSurgery (CNS) Lab, Faculty of Medicine, Health and Human Sciences, Macquarie Medical School, Macquarie University, Sydney, NSW, Australia*

## Abstract

Eye gaze tracking has traditionally employed a camera to capture a participant's eye movements and characterise their visual fixations. However, gaze pattern recognition is still challenging. This is due to both gaze point sparsity, and a seemingly random approach participants take to viewing unfamiliar stimuli without a set task.

Our paper proposes a method for integrating eye gaze into machine learning by converting a fixation's two dimensional (x, y) coordinate into a one dimensional Hilbert curve distance metric, making it well suited for implementation into machine learning. We will compare this approach to a traditional grid-based string substitution technique, with an example implementation demonstrated in a Support Vector Machine and Convolutional Neural Network. Finally, a comparison will be made to examine what method performs better.

Results have shown that this method can be both useful to dynamically quantise scanpaths for tuning statistical significance in large datasets, and to investigate the nuances of similarity found in shared bottom-up processing when participants observe unfamiliar stimuli in a free viewing experiment. Real world applications can include expertise-related eye gaze prediction, medical screening, and image saliency identification.

**Keywords:** Neuroscience, eye tracking, fractals, support vector machine, convolutional neural network.

## 1. Introduction

The development of medical devices in the past several decades has introduced various imaging modalities including ultrasound, computed tomography, magnetic resonance imaging, and others. This has led to a need for medical image analysis of properties including surface orientations and scenic depth attributes within the texture of these scans (Lopes and Betrouni, 2009), providing additional understanding of the complex patterns related to pathology.

Since the description of a participant's visual gaze pattern as a 'scanpath' by Noton and Stark (1971), a need has emerged to find physiological factors that drive the way we perceive the world. Indeed, this has now led to increasing interest into how eye gaze patterns may help to characterise the cognitive function of experts (Suman et al., 2021b), or predict and diagnose brain disease (Liu et al., 2021).

However, typical eye gaze patterns are spatially complex, and adding more dimensions (e.g., time) can give rise to the 'Curse of Dimensionality' (Bellman and Bellman, 1961) where

higher dimensional spaces grow so fast that data becomes sparse. Traditional methods for reducing dimensionality of human gaze patterns include string substitution methods, where regions of interest or evenly spaced grids over a stimulus quantise fixations into 'areas of interest' (AOI). Indeed, developing a framework for quantising human gaze into features suitable for a trained model in a neural network has been a challenge.

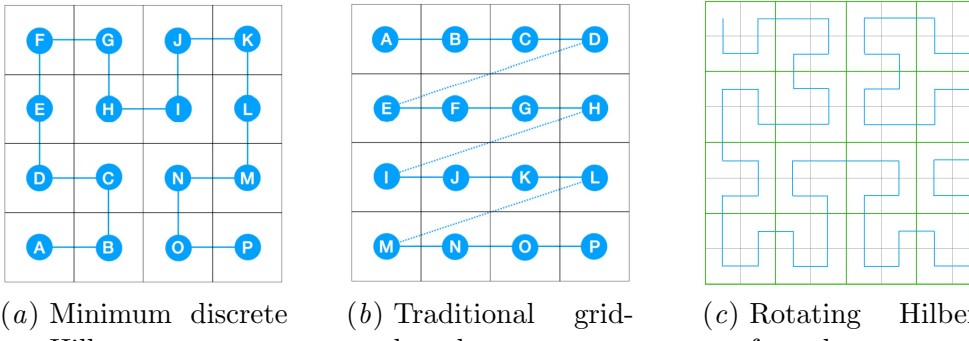

(a) Minimum discrete Hilbert curve.

(b) Traditional grid-based curve.

(c) Rotating Hilbert fractal.

Figure 1: A 4 pixel by 4 pixel image space (left, middle) converted to 1D coordinate space using a Hilbert (left) and grid-based (middle) path. As resolution increases from 4 pixel by 4 pixel (left, middle) to 8 pixel by 8 pixel space (right) the shape assumes a fractal 'self similar' pattern.

This paper presents an approach developed over a century (Peano, 1890) ago to map coordinates between 2D and 1D space while adequately preserving locality. These 1D coordinates on a 'curve' move through all locations on a unit square in a unique pattern. A simple way to understand this is illustrated in Figure 1(b) where the coordinates "snake" around the image: the first pixel is processed at the origin, and when the first line is processed, the pixel jumps to the next line and continues. In this paper we aim to present a different route than the 'snaking' pattern used in Cartesian coordinate processing and provide a demonstration of how it can be used to create tensors as feature spaces for machine learning.

## 2. Fractal Curves

Over a century ago Peano (1890) first described a way to continuously map a curve on a unit interval that could pass through every point on a unit square theoretically in infinite space. The full range of these curves span entire $n$-dimensional hypercubes of Euclidean space without endpoints. It is now known that structures like the Peano's curve can be used as frameworks in machine learning algorithms like $k$-Nearest Neighbour, where multidimensional points in a hypercube clustered on a space filling curve can define a feature space (Cover and Hart, 1967).

Peano's work paved the way for David Hilbert's discovery of a new type of fractal space curve (Hilbert, 1891) used frequently today in applications requiring better locality preservation when moving between 2D and 1D space. For example, a Euclidean point

converted to a 1D Hilbert curve can be graphed against time for a clearer empirical analysis than if it were left in 2D or 3D space. Furthermore, normalisation of multidimensional data in a 1D space would lead to increased precision as pixel resolution increases, without the need for a linear piecewise function used to interpret grid changes used in other dimensionality reduction techniques, such as string edit methods. This is because the curve can preserve locality well due to its homeomorphic exponential growth with the $n$th approximation to the limiting curve. Thus, as it increases in size, its ratio of detail to scale remains constant.

This example of the Hilbert Curve's regularity of self-similarity is a testament to its robust ability to preserve locality during a shape's growth or change during variable quantisation (v. Neumann, 1930). This makes it particularly robust in an application where multidimensional values require both reduction to one dimension and the capability for dynamic quantisation.

## 3. Tensors

Tensors are frequently used in dynamic systems and can be found as critical components for the modelling of quantum system dynamics (Strathearn, 2020) to simpler single layer neural networks demonstrated in Section 5 Method of this paper. But at their simplest, tensors can be easily grasped as a function of a specific data structure (Goodfellow, 2016 - 2016). For example, a zero order tensor is represented by a scalar, a first order tensor by a vector (or one dimensional array) and a second order tensor by a matrix (or two dimensional array). Tensors can grow to be very complex multidimensional arrays of functions that include special operations, e.g., the Hadamard product, where two tensors of the same size create a third tensor resulting from an element-wise multiplication of each parent scalar.

Tensors are an instrumental component in machine learning models, such as convolutional neural networks. However, a challenge exists in adapting things in the real world into mathematical tensor structures. We have come a long way since Russell Kirsch used a 176 x 176 grid of numbers to represent a photo of his son in 1957, captured by what is considered to be the first scanner. Those grids of numbers came to be known as 'picture elements,' first shortened to *pixels* by JPL's Billingsley (1966). During this period, work by Hubel and Wiesel (1959) on cat brains began the discovery of what we understand to be bottom up processing today. Indeed, this was the first step to implementing image decomposition layers used in convolutional neural networks today.

However, representing eye gaze patterns, termed 'scanpaths' by Noton and Stark (1971), remains a challenge due to several factors. Firstly, scanpaths are very sparse, especially when the capture duration is short. This leads to a challenge in both quantisation, to increase statistical significance, and also in the training set size required to find suitable clustering boundaries. This can be exacerbated when the stimulus is unfamiliar, and the subject is allowed to view it freely with no assigned task. The cause of this can be attributed to the nature of bottom up processing; healthy people assess an unfamiliar stimulus by examining edges and general shapes, much like the cat brains studied by Hubel and Wiesel (1959). Secondly, a flat representation of lingering (x,y) scanpath observations, known as 'fixations' by Noton and Stark (1971), leaves out crucial temporal aspects of the scanpath. These 'jerk like movements', called 'saccades' by French ophthalmologist Javal (1878), hint at important behavioural characteristics which should be included in any eye tracking study.

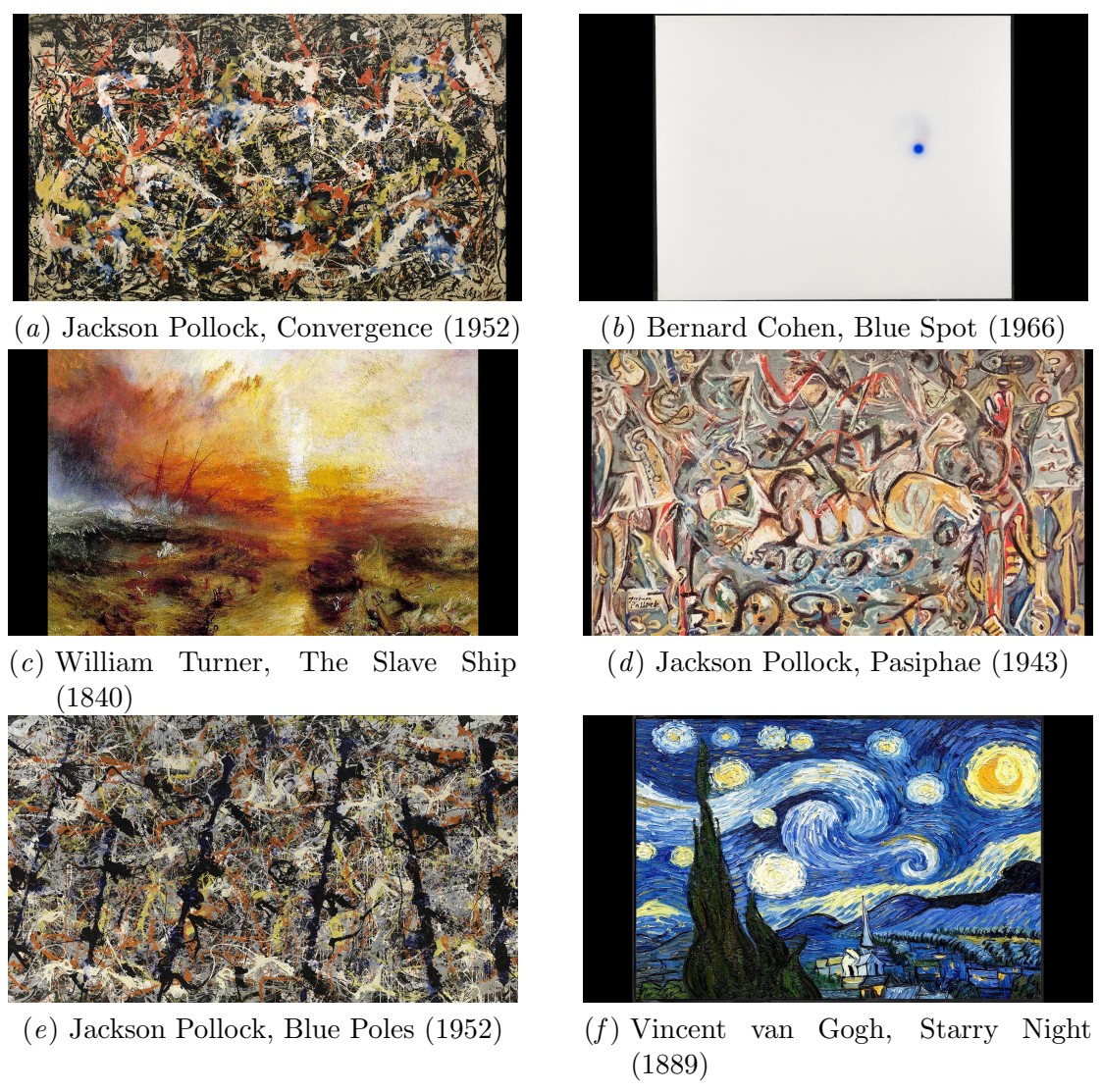

(*a*) Jackson Pollock, Convergence (1952)

(*b*) Bernard Cohen, Blue Spot (1966)

(*c*) William Turner, The Slave Ship (1840)

(*d*) Jackson Pollock, Pasiphae (1943)

(*e*) Jackson Pollock, Blue Poles (1952)

(*f*) Vincent van Gogh, Starry Night (1889)

Figure 2: Artworks possessing various levels of abstraction and extremes in geometric complexity, e.g., Pollock's paintings being complex and Cohen's being comparatively simple.

Thirdly, human vision is prone to anisotrophic effects, where some visual behaviours can be quite different in varying circumstances. Human vision is not random or chaotic, and shared characteristics like expertise (Suman et al., 2021a) and even outliers (Newport et al., 2021) due to distraction or experiment failure can be measured with mathematical frameworks that measure geometric complexity. Therefore, scanpaths are uniquely sensitive to quantisation and normalisation, due to information that can be lost, or created, as a result of both. For this reason, any scanpath representation method that uses quantisation to mitigate sparsity, or normalisation to equalise durations in eye tracking experiments, can be adversely affected.

## 4. Stimuli and Participants

Experiments were approved through The Faculty Ethics Subcommittees at Macquarie University (Sydney, Australia) and conducted by a trained Macquarie University researcher in accordance with the Australian National Statement on Ethical Conduct in Human Research. Fifty three participants labelled P01 through to P53 include medical professionals, all confirmed to be healthy at the time of the trial. No personally identifiable information can be found in the data set. The participants were enrolled in an eye-tracking machine vision study in which medical and non-medical images (e.g., paintings in this study) were used.

A noise image preceded and followed exposure to the stimulus. Multiple images were examined by participants, including six digital reproductions of artworks illustrated in Figure 2. The noise and stimulus images were scaled to fill 1920 pixels wide by 1080 pixels tall while maintaining their correct aspect ratio. Participant gaze was captured by the Eye-Link® 1000 Plus (SR Research, Ottawa, Ontario, Canada) eye-tracker operating at 1000 Hz at 0.05°Root Mean Square (RMS) and 0.25°saccade resolution.

Only one eye (right) was used for computation, even though both eyes were captured, to maximise tracking accuracy, as explained by research conducted by Hooge et al. (2018) in their paper "Gaze tracking accuracy in humans: One eye is sometimes better than two." He demonstrated that a reduction in systematic error in computed measurements could be achieved when using only one eye, if an experiment is not reliant on binocular dynamics. No chin rest or head mounting was used, resulting in the participant being 'free-to-move.' Fixations from both eyes were saved into a matrix consisting of the trial number, participant ID, eye fixations, saccades, and blinks, and a timestamp for each captured event. Post processing was used to reduce unwanted data resulting in a matrix of multiple columns including right eye $(x, y)$ coordinates and its position converted to a Hilbert distance, for each fixation.

## 5. Method

This method converts a scanpath into a Hilbert feature space using three steps. First, each 2D (x,y) fixation is converted to a 1D Hilbert distance $(h)$. Second, all Hilbert distances are quantised $(Q \cdot h)$, to reduce sparsity. Finally, $n$ number of Boolean features are initialised to zero per feature space $F^n$, with the index of $F_i$ activated if it is equal to $Q \cdot h$. A similar approach will be used for a grid-representation method. First, each 2D (x,y) fixation will

be quantised ($Q \cdot (x, y)$). Second, a grid position will be determined by summing $Q \cdot x$ to the row size multiplied by ($Q \cdot (y - 1)$). Finally, just as we did with the Hilbert method, a number of Boolean features ($F^n$) representing the feature space are initialised as zero, with the index of $F_i$ changed to 1 if it is equal to $Q \cdot h$. This process is explained using Equation 1 below:

$$\text{Let } n \text{ be number of scanpath fixations, } Q \text{ be quantisation,}$$
$$H(x, y) \text{ be Hilbert distance function, and } h = H(x, y)$$
$$\sum_{i=1}^{n} Q \cdot H(x_i, y_i) = h_i$$

$$(1)$$

$$F \text{ be feature space, indices } \in \{h\} = 1$$
$$\sum_{i=1}^{Q} F_i = \begin{cases} 1 & \text{if } i \in \{h\}; \\ 0 & \text{otherwise.} \end{cases}$$

### 5.1. Hilbert distance calculation

The strength of the Hilbert curve's locality preservation and the peculiarity of its self-similar rotating replication causes the conversion of 2D ($x, y$) to 1D ($h$) to be complex. Additionally, the perimeter of a Hilbert curve extends infinitely outward, requiring an $N$ by $N$ space to define its boundary before a position inside of it can be defined. The following pseudo-code Wikipedia (2022) demonstrates how ($x, y$) are rotated (visually illustrated in Figure 1($c$)) based on iterative changes to $rx$ and $ry$ as the curve moves from upper to lower positions:

```
int xy2d (int n, int x, int y) {
    int rx, ry, s, d=0;
    for (s=n/2; s>0; s/=2) {
        rx = (x & s) > 0;
        ry = (y & s) > 0;
        d += s * s * ((3 * rx) ^ ry);
        if (ry == 0) {
            if (rx == 1) {
                x = n-1 - x;
                y = n-1 - y;
            }
            int t  = x;
            x = y;
            y = t;
        }
    }
    return d;
}
```

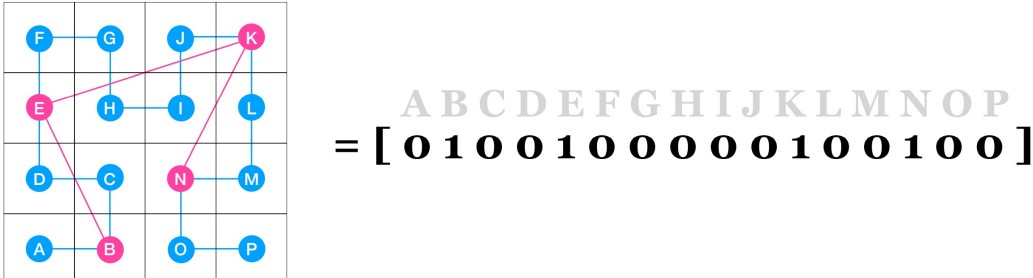

Figure 3: A 4 pixel by 4 pixel image space. 2D fixation points in pink (B, E, K, N) are activated in the boolean 1D 16-unit block representing the feature space (right).

### 5.2. Quantisation Parameters

Hilbert quantisation was empirically set to $Q, F_n = 256$. Each feature space $F$ with size $Q$ was pre-allocated with zeros defining each index. Each fixation's $h(x, y)$ result is rounded to the nearest integer, representing an 'activation index' of bins in feature space $F$, as demonstrated in Figure 3. Traditional grid-based quantisation is approached the same way, with the goal of dividing the stimulus area into 256 total grids. This results in a 16 by 16 grid, each comprised of 120 pixel by 67.5 pixel areas of interest (AOI). Fixations falling into this AOI are replaced with their bin representation similarly to the Hilbert method, where the index of the feature space $F$ is 'activated' or assigned a value of 1, if a quantised fixation equals feature space index $F_i$.

### 5.3. Feature Space

In this experiment, we created a feature space the same size as our quantisation amount $(Q, F^n = 256)$, so that each quantised segment, represented as a Boolean feature, was activated (i.e., changed from 0 to 1) if a scanpath fixation fell within it. With 53 participants, this means each SVM training session contained 52 (removing one sample for testing) samples, each containing 256 features. CNN multi-class prediction used all participant trials labelled with their corresponding stimulus exposures. SVM binary prediction was conducted between all stimuli in an A versus B manner, without repetition, independent of order. This resulted in 15 combinations predicting Painting A versus Painting B.

### 5.4. Support Vector Machine (SVM)

The driving principal behind SVM is to divide a group for the purpose of classification via 'support vectors.' A support vector defines the border that separates two prediction groups (Cristianini and Shawe-Taylor, 2000). This border is referred to as a hyperplane, rather than a line, due to its ability to be multidimensional. For example, if we have a data set of biological measurements for a large set of athletes, and we want to predict sumo wrestlers from basketball players, we can cluster them using a two dimensional height and weight feature space using a margin between the typically heavy and shorter wrestlers from the

lighter and taller basketball players. However, if we want to separate boxers from sumo wrestlers, the differential attributes between the different types of athletes may be harder to determine. For this reason, more athlete attributes (i.e., features) may be required to better separate them, e.g., arm span, leg length, etc. With greater attributes in the feature space, we now need a multidimensional way to define each athlete during classification, and a hyperplane to separate them. Matlab was used to conduct the SVM trials, using the `fitcsvm` function to create the model and the `predict` function to test, based on a conventional binary testing SVM algorithm (Cristianini and Shawe-Taylor, 2000) utilising default Matlab parameters.

### 5.4.1. Testing and Validation

Testing and validation will be done using a 'Leave-One-Out Cross-Validation' approach (Sammut and Webb, 2010). Much like its name implies, this method requires that each sample in the data set is iteratively excluded from training, to be used as a prediction test sample. At completion, all prediction results are averaged for mean accuracy. In each of our 15 tests, the 53 scanpaths in Stimulus A and 53 in Stimulus B resulted in 106 folds during validation, meaning every scanpath instance was used during each test trial.

### 5.5. Convolutional Neural Network (CNN)

The appeal of Convolutional Neural Networks (CNN) are primarily centred around their ability to automatically extract features from large inputs, e.g., digital images, using a system of layers which distil an input matrix to an output result (Goodfellow, 2016 - 2016). This output result, called a neuron as a digital representation of its namesake, correlates to a label which reveals an inference. The 'convolutional' aspect of CNN refers to filters in the process designed to activate particular elements of the input over others. Typically, these filters use 2D kernels (e.g., 3 pixel by 3 pixel) floating over a 2D feature space (e.g., 100 pixel by 100 pixel). Other layers include a Rectified Linear Unit (ReLU) which maps negative values to zero while leaving positive values unchanged and the Softmax layer, used to negotiate activation for multi-class classification, i.e., when there are more than two labels.

In our implementation of CNN, we converted all the participant scanpaths to individual inputs, resulting in (53 participants · 6 paintings) 318 rows of 256 features. In lieu of the binary classification we used in SVM, we chose to exploit CNN's multi-class classification capabilities, requiring a label matrix corresponding to the stimulus each participant was exposed to in their trial. We used the same level of quantisation ($Q = 256$) that was used in SVM. This resulted in a Matlab image input layer with height as 256, width as 1, and channels as 1 ([256 1 1]). We added a convolution layer with a kernel size of 5x5 followed by a ReLU layer, two fully connected layers, followed by a Softmax layer and final classification layer. We conducted our training over 500 epochs of 2 iterations each.

## 6. Results

All results were computed on a 2018 Macbook Pro (2.2 GHz 6-Core Intel Core i7) with 16 GB 2400 MHz DDR4 memory, Intel UHD Graphics 630 1536 MB, running Matlab

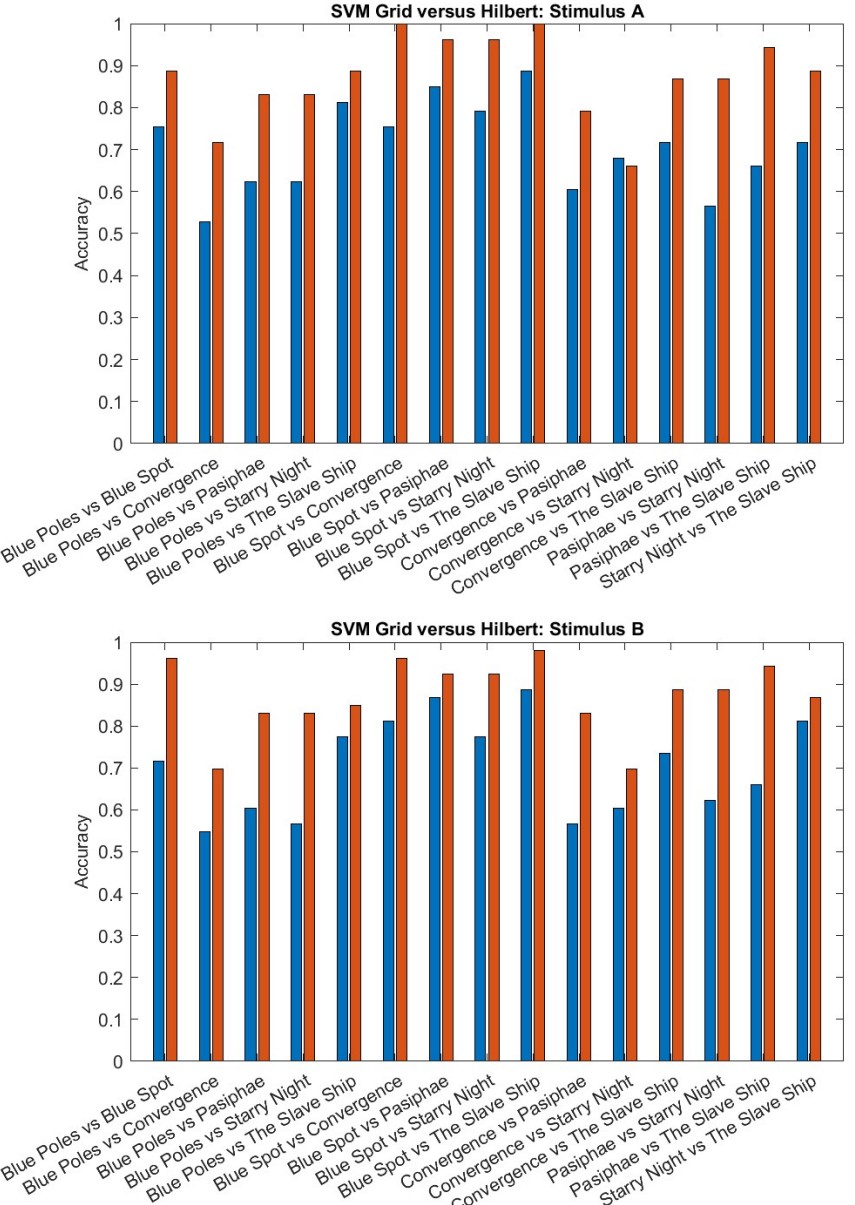

Figure 4: Support Vector Machine (SVM) accuracy comparison between Grid (blue bar on left) versus Hilbert (orange bar on right) methods. Top chart indicates accuracy for Stimulus A, or the first stimulus, in a binary prediction, e.g., 'Blue Poles' in 'Blue Poles vs Blue Spot.' Bottom chart indicates accuracy for Stimulus B.

**CNN Predictions using Hilbert**

| Truth | Blue Poles | Conv. | Blue Spot | Starry Night | Slave Ship | Pasiphae | **ACC** |
|---|---|---|---|---|---|---|---|
| Blue Poles | 48 | 1 | 0 | 2 | 2 | 0 | 0.9057 |
| Conv. | 2 | 47 | 0 | 4 | 0 | 0 | 0.8868 |
| Blue Spot | 0 | 0 | 53 | 0 | 0 | 0 | 1.0000 |
| Starry Night | 1 | 4 | 0 | 48 | 0 | 0 | 0.9057 |
| Slave Ship | 1 | 1 | 0 | 0 | 51 | 0 | 0.9623 |
| Pasiphae | 0 | 2 | 0 | 1 | 1 | 49 | 0.9245 |

Table 1: CNN predictions using the Hilbert curve illustrated in Figure 1(a) and Figure 1(c) to extract features.

**CNN Predictions using Grid**

| Truth | Blue Poles | Conv. | Blue Spot | Starry Night | Slave Ship | Pasiphae | **ACC** |
|---|---|---|---|---|---|---|---|
| Blue Poles | 46 | 3 | 1 | 1 | 0 | 2 | 0.8679 |
| Conv. | 8 | 41 | 0 | 1 | 1 | 2 | 0.7736 |
| Blue Spot | 2 | 3 | 45 | 2 | 0 | 1 | 0.8491 |
| Starry Night | 2 | 1 | 0 | 48 | 1 | 1 | 0.9057 |
| Slave Ship | 2 | 1 | 0 | 0 | 50 | 0 | 0.9434 |
| Pasiphae | 1 | 2 | 0 | 4 | 1 | 45 | 0.8491 |

Table 2: CNN predictions using the grid-based curve illustrated in Figure 1(b) to extract features.

R2022b. Performance details were not recorded, but were typically less than a few minutes for training and testing either SVM or CNN.

Overall, using a Hilbert curve instead of a traditional grid-based curve resulted in increased performance using both SVM and CNN machine learning methods. With SVM, only three grid-based methods performed better than Hilbert curves out of all 60 trials whereas with CNN, no grid-based CNN trials performed better, with Starry Night being the only stimulus performing as well as Hilbert.

## 6.1. SVM Results

All results in our SVM test shown in Figure 4, except for one, showed improved prediction accuracy using a Hilbert curve to define the 1D path through the quantised stimulus space. Notable differences in prediction accuracy can be seen in Hilbert's increased ability to discriminate between Pasiphae and Starry Night, and Pasiphae and The Slave Ship. Blue Spot also showed increased prediction ability, though this is probably due to its unique lack of geometric complexity being exploited by the Hilbert method's novel geo-spatial sensitivity.

However, Hilbert performed slightly poorer than the grid-based method when predicting for Convergence during a Convergence versus Starry Night test. Interestingly, Hilbert's performance predicting Starry Night in the same test performed adequately better than the grid-based method. Other tests where Hilbert only performed slightly better than the grid-based method include predicting The Slave Ship (Starry Night versus The Slave Ship), Pasiphae (Blue Spot versus Pasiphae), and Blue Poles (Blue Poles versus The Slave Ship).

## 6.2. CNN Results

All results in our CNN test shown in Table 1 and 2, except for one, showed improved prediction accuracy using a Hilbert curve to define the 1D path through the quantised stimulus space. Notable differences in Hilbert's increased prediction accuracy can be seen in Blue Spot's true positive rate, but also in Pasiphae's true negative rate, both of which did poorly using grid-based methods. Indeed, overall accuracy using CNN over SVM can be seen throughout the results, making Hilbert use of spatial feature extraction a viable alternative to grid-based methods in Deep Learning.

One result where Hilbert did not do better, but matched in accuracy with the grid method, was in predicting Starry Night. Indeed, this stimulus was also challenging with SVM where Hilbert did worse at predicting Convergence and only slightly better at predicting The Slave Ship when both were matched with Starry Night.

## 7. Discussion

This paper has demonstrated that implementation of a Hilbert fractal curve can be used as a 1D first order tensor in an SVM and CNN predictor data set, and that it performs better than traditional grid-based methods when dimensionality and quantisation are equal. In most cases, a large improvement is seen due to Hilbert's increased spatial sensitivity due to its unique fractal path through the stimulus space. Indeed, the performance shown with

SVM indicates that it is suitable for potential use in a clinical setting. Such uses could include screening for impairment while testing cognitive function. Impairment could include alcohol or drug intoxication, or brain injuries such as ataxia or certain forms of dementia, where eye movements are effected by damage to the brain (Liu et al., 2021). Pedagogical applications can include separating naive and expert observers, e.g., radiographers searching for cancers in medical images. If high-performing radiographers exhibit similar search patterns when examining a brain MRI, an ML model could represent a method for measuring, comparing, and training perceptual performance (Suman et al., 2021b). Future work in this area could explore how models of expert radiographers can transfer their perceptual patterns to an AI network for better saliency prediction.

However, in cases where Hilbert did not perform as well (i.e., Convergence, The Slave Ship, Pasiphae, and Blue Poles using SVM), a reason for this drop deserves some further scrutiny. In the SVM Starry Night versus The Slave Ship trial, both paintings appear to have a similar visual geometric complexity. However, The Slave Ship may lack compelling visual symbols to drive differential bottom-up percepts compared to Starry Night's painterly strokes. Indeed, a clinical assessment into the higher cognition functions that drive such percepts are outside the scope of this paper. However, empirically, The Slave Ship appears to be a lot more subdued compared to the higher contrast seen in Starry Night. Similarly, Convergence, Blue Poles, and Pasiphae all share a high degree of contrast, abstraction, and geometric complexity, which may play a role in the degree to which they scramble visual fixation patterns during bottom-up search. These challenges in separability were mostly resolved under CNN. However, Starry Night was still challenging, bringing Hilbert performance on par with the poorer performing grid-based method.

Future work should include an optimisation model used in lieu of the empirical method this paper used to find a quantisation amount ($Q$). Due to the performant nature of the 1D tensors used in this method, the optimisation model could simply use a brute force method to iterate through a range of quantisation amounts until an optimum margin is found, using that optimised $Q$ for training. Additionally, tandem progress in both computational methods to expand the feature space, and also an increased understanding of how bottom-up percepts can be integrated into an overall visual perception model should also be explored. Integrating temporal aspects of gaze would be an important inclusion into a future method, allowing for analysis of more complex visual patterns such as re-reading and inhibition of return. Furthermore, due to Hilbert's improved locality preservation during changes in resolution, quantisation can be dynamically reduced and increased in layers extracted from a Hilbert feature space, convolved using 1D kernels, in a neural network integrating the temporal dimension. All these methods demonstrate many avenues of improvement for adopting Hilbert curves as transformations for dimensionality reduction of multidimensional data sets.

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
