# OpenReview forum: "Integrating eye gaze into machine learning using fractal curves"
_NeurIPS.cc/2022/Workshop/GMML — Gaze Meets ML 2022 Poster_

### Official Review · Reviewer_uU4p · 2022-10-14
**Not particularly novel but might be useful to some workshop participants**

**Rating:** 6
**Confidence:** 3

**Review:**

The motivation and methods of this paper are oddly written, although it appears that most of the necessary details are in the manuscript somewhere. The proposal is to simply replace a grid-based quantization of an image viewed by humans with a quantization based on Hilbert curves. The authors then show that representing the same eye gaze fixation data (collected on images such as art pieces) using these two different methods produces different classification results. They tested both a series of binary SVM classification tasks (e.g. classifying image1 vs image2, image1 vs. image 3...) as well as a multi-class task using a CNN. In most cases, the Hilbert curve method produced higher classification accuracy. To me this paper is quite borderline -- I do not see any glaring flaws that would warrant outright rejection, but the work does not seem particularly novel or insightful. However, some researchers might find the idea useful for their own work and might appreciate the contribution at the workshop. I will admit that I'm not particularly well-read on different applications of machine learning to gaze data, so I could be missing some motivation from this method by simply relying on the authors' text.

As for novelty -- using Hilbert curve features is not particularly new in machine learning, as the authors point out themselves with a citation to some work from 1967. More recently it has been used in studies of machine learning on other human data (e.g.  https://ieeexplore.ieee.org/abstract/document/8787290/authors#authors; https://link.springer.com/chapter/10.1007/978-3-642-39094-4_17). One work even applies Hilbert curve quantization to eyetracking scanpaths to identify outliers, which the authors even cite (citation 12).

This paper also does not provide any particular insights beyond the idea that "higher spatial resolution in input features is useful." One of the motivations they give in the Introduction is that high-dimensional data is difficult to represent -- however, here they are not representing a new dimension in the data (e.g. time) compared to the grid-based method. They're simply changing the spatial resolution. This is ultimately fine, but a small advance.

*Miscellaneous suggestions/questions*
* It was difficult to compare Tables 1 and 2. Please combine the accuracies for Hilbert vs. grid into one table. You could use heatmaps to represent the classification predictions if you want to keep that information in the paper.
* A proper double-blinding of the work would omit the name of the university in the description of the dataset.
* What types of medical images were used in the dataset? If there's another publication which used the data, it should be cited, or if the data are shared publicly it should be linked.
* Please add some estimates of variance across the classification cross-validation folds, e.g. standard deviation or confidence intervals.
* Not sure why the authors devoted section 3 to explaining tensors. If they were going to spend time giving basic background, I think it would be more appropriate to go into a bit of detail about the grid-based string substitution methods that are the comparison condition.
* The last paragraph of section 3 seems particularly out of place, given that it's not about tensors. I don't think it's even useful to talk about saccades and gaze duration at all before the discussion -- in the paper the authors are ignoring gaze duration and treating all fixation points similarly regardless of their duration, and refer to "integrating temporal aspects of gaze" as a future direction (line 278).

---

### Official Review · Reviewer_CeCN · 2022-10-19
**This paper explores benefits to encoding eye gaze using hilbert curves as opposed to using grid-based curves in machine learning algorithms (SVM and CNN) in an image free viewing context. The paper is easy to follow and the study is well-motivated. The paper presents insights into how visual gaze pattern or scan path can be better incorporated into training of machine learning models through using Hilbert curves.**

**Rating:** 7
**Confidence:** 4

**Review:**

Quality: Medium; Clarity: High; Originality: Medium; Significance: Medium

Pros:
1) Problem well-motivated
2) Detailed methods
3) Good discussion of implication of findings and future work

Cons:
1) No chin rest or head mounting was used in the study. What measures were taken to make sure that the participants did not move and that their eye gaze can be accurately tracked? How was the gaze tracker calibrated? Did any experimental trials need to be omitted?
2) How long on average did users spend observing each image? Was the time controlled or was the users allowed to move-on to the next image freely.
3) Unnecessary background in Section 5.4 and Section 5.5 on Support Vector Machine and Convolutional Neural Network.
4) Combine table 1 and table 2 for easier comparisons

---

### Official Review · Reviewer_P4se · 2022-10-19
**Interesting work**

**Rating:** 8
**Confidence:** 4

**Review:**

The paper proposes the use of a Hilbert fractal curve to model eye gaze tracking in modern user interaction devices.
The argument of using the Hilbert fractal curve instead of a traditional grid based method is that is makes the data more suitable for downstream machine learning task.
The paper presents a convincing idea and it is well written.
The preliminary study with 53 participants is enough material for a workshop paper.

---

### Meta-Review · Area_Chair_ySRo · 2022-10-20

**Recommendation:** Accept (Poster)
**Confidence:** 4

**Metareview:**

The work proposes converting the scan path fixations from two-dimensional (x,y) information to a one-dimensional Hilbert space fractal distance metric curve. The authors compare this with traditional grid-based string substitution techniques and demonstrate the results using two machine learning techniques - SVM and CNNs.

The reviews found it interesting and had concrete suggestions on how to improve the paper in terms of its readability. Overall I recommend an accept and think this will make for an interesting discussion in the workshop.

---

### Decision · Program_Chairs · 2022-10-20

Accept (Poster)